# $M$-Statistic for Kernel Change-Point Detection

**Shuang Li, Yao Xie**
H. Milton Stewart School of
Industrial and Systems Engineering
Georgian Institute of Technology
sli370@gatech.edu
yao.xie@isye.gatech.edu

**Hanjun Dai, Le Song**
Computational Science and Engineering
College of Computing
Georgia Institute of Technology
hanjundai@gatech.edu
lsong@cc.gatech.edu

## Abstract

Detecting the emergence of an abrupt change-point is a classic problem in statistics and machine learning. Kernel-based nonparametric statistics have been proposed for this task which make fewer assumptions on the distributions than traditional parametric approach. However, none of the existing kernel statistics has provided a computationally efficient way to characterize the extremal behavior of the statistic. Such characterization is crucial for setting the detection threshold, to control the significance level in the offline case as well as the average run length in the online case. In this paper we propose two related computationally efficient $M$-statistics for kernel-based change-point detection when the amount of background data is large. A novel theoretical result of the paper is the characterization of the tail probability of these statistics using a new technique based on change-of-measure. Such characterization provides us accurate detection thresholds for both offline and online cases in computationally efficient manner, without the need to resort to the more expensive simulations such as bootstrapping. We show that our methods perform well in both synthetic and real world data.

## 1 Introduction

Detecting the emergence of abrupt change-points is a classic problem in statistics and machine learning. Given a sequence of samples, $x_1, x_2, \ldots, x_t$, from a domain $\mathcal{X}$, we are interested in detecting a possible change-point $\tau$, such that before $\tau$, the samples $x_i \sim P$ *i.i.d.* for $i \leq \tau$, where $P$ is the so-called background distribution, and after the change-point, the samples $x_i \sim Q$ *i.i.d.* for $i \geq \tau+1$, where $Q$ is a post-change distribution. Here the time horizon $t$ can be either a fixed number $t = T_0$ (called an offline or fixed-sample problem), or $t$ is not fixed and we keep getting new samples (called a sequential or online problem). Our goal is to detect the existence of the change-point in the offline setting, or detect the emergence of a change-point as soon as possible after it occurs in the online setting. We will restrict our attention to detecting one change-point, which arises often in monitoring problems. One such example is the seismic event detection [9], where we would like to detect the onset of the event precisely in retrospect to better understand earthquakes or as quickly as possible from the streaming data. Ideally, the detection algorithm can also be robust to distributional assumptions as we wish to detect all kinds of seismic events that are different from the background. Typically we have a large amount of background data (since seismic events are rare), and we want the algorithm to exploit these data while being computationally efficient.

Classical approaches for change-point detection are usually parametric, meaning that they rely on strong assumptions on the distribution. Nonparametric and kernel approaches are distribution free and more robust as they provide consistent results over larger classes of data distributions (they can possibly be less powerful in settings where a clear distributional assumption can be made). In particular, many kernel based statistics have been proposed in the machine learning literature [5, 2, 18, 6, 7, 1] which typically work better in real data with few assumptions. However, none of these existing kernel statistics has provided a computationally efficient way to characterize

the tail probability of the extremal value of these statistics. Characterization such tail probability is crucial for setting the correct detection thresholds for both the offline and online cases. Furthermore, efficiency is also an important consideration since typically the amount of background data is very large. In this case, one has the freedom to restructure and sample the background data during the statistical design to gain computational efficiency. On the other hand, change-point detection problems are related to the statistical two-sample test problems; however, they are usually more difficult in that *for change-point detection, we need to search for the unknown change-point location $\tau$*. For instance, in the offline case, this corresponds to taking a maximum of a series of statistics each corresponding to one putative change-point location (a similar idea was used in [5] for the offline case), and in the online case, we have to characterize the average run length of the test statistic hitting the threshold, which necessarily results in taking a maximum of the statistics over time. Moreover, the statistics being maxed over are usually highly correlated. Hence, analyzing the tail probabilities of the test statistic for change-point detection typically requires more sophisticated probabilistic tools.

In this paper, we design two related $M$-statistics for change-point detection based on kernel maximum mean discrepancy (MMD) for two-sample test [3, 4]. Although MMD has a nice unbiased and minimum variance $U$-statistic estimator (MMD$_u$), it can not be directly applied since MMD$_u$ costs $\mathcal{O}(n^2)$ to compute based on a sample of $n$ data points. In the change-point detection case, this translates to a complexity quadratically grows with the number of background observations and the detection time horizon $t$. Therefore, we adopt a strategy inspired by the recently developed $B$-test statistic [17] and design a $\mathcal{O}(n)$ statistic for change-point detection. At a high level, our methods sample $N$ blocks of background data of size $B$, compute quadratic-time MMD$_u$ of each reference block with the post-change block, and then average the results. However, different from the simple two-sample test case, in order to provide an accurate change-point detection threshold, the background block needs to be designed in a novel structured way in the offline setting and updated recursively in the online setting.

Besides presenting the new $M$-statistics, our contributions also include: (1) deriving accurate approximations to the significance level in the offline case, and average run length in the online case, for our $M$-statistics, which enable us to determine thresholds efficiently without recurring to the onerous simulations (e.g. repeated bootstrapping); (2) obtaining a closed-form variance estimator which allows us to form the $M$-statistic easily; (3) developing novel structured ways to design background blocks in the offline setting and rules for update in the online setting, which also leads to desired correlation structures of our statistics that enable accurate approximations for tail probability. To approximate the asymptotic tail probabilities, we adopt a highly sophisticated technique based on change-of-measure, recently developed in a series of paper by Yakir and Siegmund et al. [16]. The numerical accuracy of our approximations are validated by numerical examples. We demonstrate the good performance of our method using real speech and human activity data. We also find that, in the two-sample testing scenario, it is always beneficial to increase the block size $B$ as the distribution for the statistic under the null and the alternative will be better separated; however, this is no longer the case in online change-point detection, because a larger block size inevitably causes a larger detection delay. Finally, we point to future directions to relax our Gaussian approximation and correct for the skewness of the kernel-based statistics.

## 2 Background and Related Work

We briefly review kernel-based methods and the maximum mean discrepancy. A *reproducing kernel Hilbert space (RKHS)* $\mathcal{F}$ on $\mathcal{X}$ with a kernel $k(x, x')$ is a Hilbert space of functions $f(\cdot): \mathcal{X} \mapsto \mathbb{R}$ with inner product $\langle \cdot, \cdot \rangle_{\mathcal{F}}$. Its element $k(x, \cdot)$ satisfies the reproducing property: $\langle f(\cdot), k(x, \cdot) \rangle_{\mathcal{F}} = f(x)$, and consequently, $\langle k(x, \cdot), k(x', \cdot) \rangle_{\mathcal{F}} = k(x, x')$, meaning that we can view the evaluation of a function $f$ at any point $x \in \mathcal{X}$ as an inner product.

Assume there are two sets with $n$ observations from a domain $\mathcal{X}$, where $X = \{x_1, x_2, \ldots, x_n\}$ are drawn *i.i.d.* from distribution $P$, and $Y = \{y_1, y_2, \ldots, y_n\}$ are drawn *i.i.d.* from distribution $Q$. The *maximum mean discrepancy* (MMD) is defined as [3] $\text{MMD}_0[\mathcal{F}, P, Q] := \sup_{f \in \mathcal{F}} \{\mathbb{E}_x[f(x)] - \mathbb{E}_y[f(y)]\}$. An unbiased estimate of $\text{MMD}_0^2$ can be obtained using $U$-statistic

$$\text{MMD}_u^2[\mathcal{F}, X, Y] = \frac{1}{n(n-1)} \sum_{i,j=1, i \neq j}^{n} h(x_i, x_j, y_i, y_j), \qquad (1)$$

where $h(\cdot)$ is the kernel of the $U$-statistic defined as $h(x_i, x_j, y_i, y_j) = k(x_i, x_j) + k(y_i, y_j) - k(x_i, y_j) - k(x_j, y_i)$. Intuitively, the empirical test statistic $\mathrm{MMD}_u^2$ is expected to be small (close to zero) if $P = Q$, and large if $P$ and $Q$ are far apart. The complexity for evaluating (1) is $O(n^2)$ since we have to form the so-called Gram matrix for the data. Under $H_0$ ($P = Q$), the $U$-statistic is degenerate and distributed the same as an infinite sum of Chi-square variables.

To improve the computational efficiency and obtain an easy-to-compute threshold for hypothesis testing, recently, [17] proposed an alternative statistic for $\mathrm{MMD}_0^2$ called $B$-test. The key idea of the approach is to partition the $n$ samples from $P$ and $Q$ into $N$ non-overlapping blocks, $X_1, \ldots, X_N$ and $Y_1, \ldots, Y_N$, each of constant size $B$. Then $\mathrm{MMD}_u^2[\mathcal{F}, X_i, Y_i]$ is computed for each pair of blocks and averaged over the $N$ blocks to result in $\mathrm{MMD}_B^2[\mathcal{F}, X, Y] = \frac{1}{N} \sum_{i=1}^{N} \mathrm{MMD}_u^2[\mathcal{F}, X_i, Y_i]$. Since $B$ is constant, $N \sim O(n)$, and the computational complexity of $\mathrm{MMD}_B^2[\mathcal{F}, X, Y]$ is $O(B^2 n)$, a significant reduction compared to $\mathrm{MMD}_u^2[\mathcal{F}, X, Y]$. Furthermore, by averaging $\mathrm{MMD}_u^2[\mathcal{F}, X_i, Y_i]$ over independent blocks, the $B$-statistic is asymptotically normal leveraging over the central limit theorem. This latter property also allows a simple threshold to be derived for the two-sample test rather than resorting to more expensive bootstrapping approach. Our later statistics are inspired by $B$-statistic. However, the change-point detection setting requires significant new derivations to obtain the test threshold since one cares about the maximum of $\mathrm{MMD}_B^2[\mathcal{F}, X, Y]$ computed at different point in time. Moreover, the change-point detection case consists of a sum of *highly correlated* MMD statistics, because these $\mathrm{MMD}_B^2$ are formed with a common test block of data. This is inevitable in our change-point detection problems because test data is much less than the reference data. Hence, we cannot use the central limit theorem (even a martingale version), but have to adopt the aforementioned change-of-measure approach.

**Related work.** Other nonparametric change-point detection approach has been proposed in the literature. In the offline setting, [5] designs a kernel-based test statistic, based on a so-called running maximum partition strategy to test for the presence of a change-point; [18] studies a related problem in which there are $s$ anomalous sequences out of $n$ sequences to be detected and they construct a test statistic using MMD. In the online setting, [6] presents a meta-algorithm that compares data in some "reference window" to the data in the current window, using some empirical distance measures (not kernel-based); [1] detects abrupt changes by comparing two sets of descriptors extracted online from the signal at each time instant: the immediate past set and the immediate future set; based on soft margin single-class support vector machine (SVM), they build a dissimilarity measure (which is asymptotically equivalent to the Fisher ratio in the Gaussian case) in the feature space between those sets without estimating densities as an intermediate step; [7] uses a density-ratio estimation to detect change-point, and models the density-ratio using a non-parametric Gaussian kernel model, whose parameters are updated online through stochastic gradient decent. The above work lack theoretical analysis for the extremal behavior of the statistics or average run length.

## 3  $M$-statistic for offline and online change-point detection

Give a sequence of observations $\{\ldots, x_{-2}, x_{-1}, x_0, x_1, \ldots, x_t\}$, $x_i \in \mathcal{X}$, with $\{\ldots, x_{-2}, x_{-1}, x_0\}$ denoting the sequence of background (or reference) data. Assume a large amount of reference data is available. Our goal is to detect the *existence* of a change-point $\tau$, such that before the change-point, samples are *i.i.d.* with a distribution $P$, and after the change-point, samples are *i.i.d.* with a different distribution $Q$. The location $\tau$ where the change-point occurs is unknown. We may formulate this problem as a hypothesis test, where the null hypothesis states that there is no change-point, and the alternative hypothesis is that there exists a change-point at some time $\tau$. We will construct our kernel-based $M$-statistic using the maximum mean discrepancy (MMD) to measure the difference between distributions of the reference and the test data.

We denote by $Y$ the block of data which potentially contains a change-point (also referred to as the post-change block or test block). **In the offline setting**, we assume the size of $Y$ can be up to $B_{\max}$, and we want to search for a location of the change-point $\tau$ within $Y$ such that observations after $\tau$ are from a different distribution. Inspired by the idea of $B$-test [17], we sample $N$ reference blocks of size $B_{\max}$ independently from the reference pool, and index them as $X_i^{B_{\max}}$, $i = 1, \ldots, N$. Since we search for a location $B$ ($2 \leq B \leq B_{\max}$) within $Y$ for a change-point, we construct sub-block from $Y$ by taking $B$ contiguous data points, and denote them as $Y^B$. To form the statistic, we correspondingly construct sub-blocks from each reference block by taking $B$ contiguous data points out of that block, and index these sub-blocks as $X_i^{(B)}$ (illustrated in Fig. 1(a)). We then compute

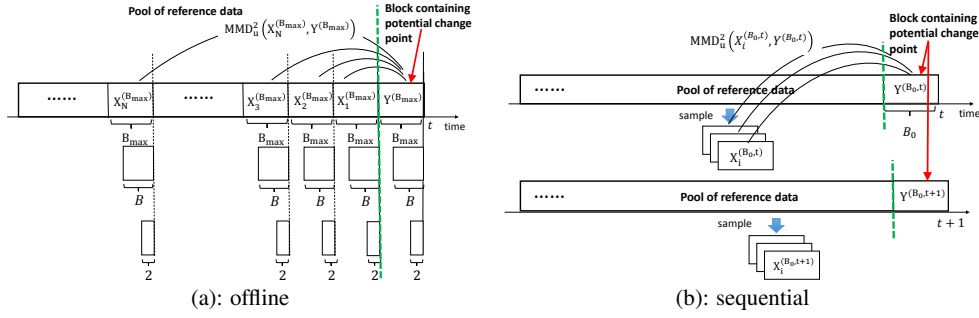

$$\text{(a): offline} \qquad\qquad\qquad \text{(b): sequential}$$

Figure 1: Illustration of (a) offline case: data are split into blocks of size $B_{\max}$, indexed backwards from time $t$, and we consider blocks of size $B$, $B = 2, \ldots, B_{\max}$; (b) online case. Assuming we have large amount of reference or background data that follows the null distribution.

$\text{MMD}_u^2$ between $(X_i^{(B)}, Y^{(B)})$, and average over blocks

$$Z_B := \frac{1}{N} \sum_{i=1}^{N} \text{MMD}_u^2(X_i^{(B)}, Y^{(B)}) = \frac{1}{NB(B-1)} \sum_{i=1}^{N} \sum_{j,l=1, j\neq l}^{B} h(X_{i,j}^{(B)}, X_{i,l}^{(B)}, Y_j^{(B)}, Y_l^{(B)}), \quad (2)$$

where $X_{i,j}^B$ denotes the $j$th sample in $X_i^{(B)}$, and $Y_j^{(B)}$ denotes the $j$ th sample in $Y^B$. Due to the property of $\text{MMD}_u^2$, under the null hypothesis, $\mathbb{E}[Z_B] = 0$. Let $\text{Var}[Z_B]$ denote the variance of $Z_B$ under the null. The expression of $Z_B$ is given by (6) in the following section. We see the variance depends on the block size $B$ and the number of blocks $N$. As $B$ increases $\text{Var}[Z_B]$ decreases (also illustrated in Figure 5 in the appendix). Considering this, we standardize the statistic, maximize over all values of $B$ to define the *offline $M$-statistic*, and detect a change-point whenever the $M$-statistic exceeds the threshold $b > 0$:

$$M := \max_{B\in\{2,3,\ldots,B_{\max}\}} \underbrace{Z_B/\sqrt{\text{Var}[Z_B]}}_{Z_B'} > b, \quad \{\text{offline change-point detection}\} \qquad (3)$$

where varying the block-size from 2 to $B_{\max}$ corresponds to searching for the unknown change-point location. **In the online setting**, suppose the post-change block $Y$ has size $B_0$ and we construct it using a sliding window. In this case, the potential change-point is declared as the end of each block $Y$. To form the statistic, we take $NB_0$ samples without replacement (since we assume the reference data are *i.i.d.* with distribution $P$) from the reference pool to form $N$ reference blocks, compute the quadratic $\text{MMD}_u^2$ statistics between each reference block and the post-change block, and then average them. When there is a new sample (time moves from $t$ to $t + 1$), we append the new sample in the reference block, remove the oldest sample from the post-change block, and move it to the reference pool. The reference blocks are also updated accordingly: the end point of each reference block is moved to the reference pool, and a new point is sampled and appended to the front of each reference block, as shown in Fig. 1(b). Using the sliding window scheme described above, similarly, we may define an *online $M$-statistic* by forming a standardized average of the $\text{MMD}_u^2$ between the post-change block in a sliding window and the reference block:

$$Z_{B_0,t} := \frac{1}{N} \sum_{i=1}^{N} \text{MMD}_u^2(X_i^{(B_0,t)}, Y^{(B_0,t)}), \qquad (4)$$

where $B_0$ is the fixed block-size, $X_i^{(B_0,t)}$ is the $i$th reference block of size $B_0$ at time $t$, and $Y^{(B_0,t)}$ is the the post-change block of size $B_0$ at time $t$. In the online case, we have to characterize the average run length of the test statistic hitting the threshold, which necessarily results in taking a maximum of the statistics over time. The online change-point detection procedure is a stopping time, where we detect a change-point whenever the normalized $Z_{B_0,t}$ exceeds a pre-determined threshold $b > 0$:

$$T = \inf\{t : \underbrace{Z_{B_0,t}/\sqrt{\text{Var}[Z_{B_0}]}}_{M_t} > b\}. \quad \{\text{online change-point detection}\} \qquad (5)$$

Note in the online case, we actually take a maximum of the standardized statistics over time. There is a recursive way to calculate the online $M$-statistic efficiently, explained in Section A in the appendix. At the stopping time $T$, we claim that there exists a change-point. There is a tradeoff in choosing the block size $B_0$ in online setting: a small block size will incur a smaller computational cost, which may be important for the online case, and it also enables smaller detection delay for strong change-

point magnitude; however, the disadvantage of a small $B_0$ is a lower power, which corresponds to a longer detection delay when the change-point magnitude is weak (for example, the amplitude of the mean shift is small). Examples of offline and online $M$-statistics are demonstrated in Fig. 2 based on synthetic data and a segment of the real seismic signal. We see that the proposed offline $M$-statistic powerfully detects the existence of a change-point and accurately pinpoints where the change occurs; the online $M$-statistic quickly hits the threshold as soon as the change happens.

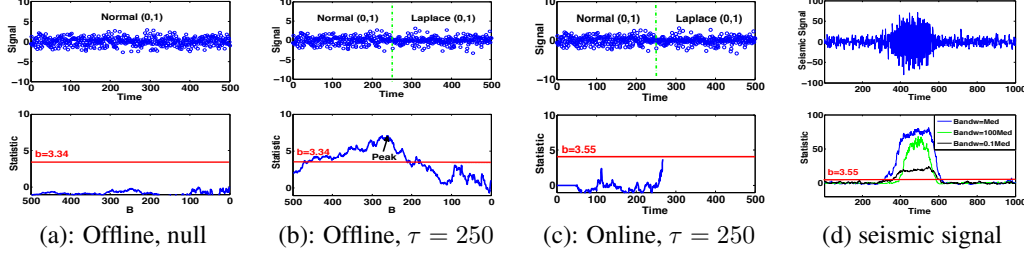

(a): Offline, null    (b): Offline, $\tau = 250$    (c): Online, $\tau = 250$    (d) seismic signal

Figure 2: Examples of offline and online $M$-statistic with $N = 5$: (a) and (b), offline case without and with a change-point ($B_{\max} = 500$ and the maximum is obtained when $B = 263$); (c) online case with a change-point at $\tau = 250$, stopping-time $T = 268$ (detection delay is 18), and we use $B_0 = 50$; (d) a real seismic signal and $M$-statistic with different kernel bandwidth. All thresholds are theoretical values and are marked in red.

## 4 Theoretical Performance Analysis

We obtain an analytical expression for the variance $\mathrm{Var}[Z_B]$ in (3) and (5), by leveraging the correspondence between the $\mathrm{MMD}_u^2$ statistics and $U$-statistic [11] (since $Z_B$ is some form of $U$-statistic), and exploiting the known properties of $U$-statistic. We also derive the covariance structure for the online and offline standardized $Z_B$ statistics, which is crucial for proving theorems 3 and 4.

**Lemma 1 (Variance of $Z_B$ under the null.)** *Given any fixed block size $B$ and number of blocks $N$, under the null hypothesis,*

$$\mathrm{Var}[Z_B] = \binom{B}{2}^{-1} \left[ \frac{1}{N} \mathbb{E}[h^2(x, x', y, y')] + \frac{N-1}{N} \mathrm{Cov}\left[h(x, x', y, y'), h(x'', x''', y, y')\right] \right], \quad (6)$$

*where $x$, $x'$, $x''$, $x'''$, $y$, and $y'$ are* i.i.d. *with the null distribution $P$.*

Lemma 1 suggests an easy way to estimate the variance $\mathrm{Var}[Z_B]$ from the reference data. To estimate (6), we need to first estimate $\mathbb{E}[h^2(x, x', y, y')]$, by each time drawing four samples without replacement from the reference data, use them for $x, x', y, y'$, evaluate the sampled function value, and then form a Monte Carlo average. Similarly, we may estimate $\mathrm{Cov}\left[h(x, x', y, y'), h(x'', x''', y, y')\right]$.

**Lemma 2 (Covariance structure of the standardized $Z_B$ statistics.)** *Under the null hypothesis, given $u$ and $v$ in $[2, B_{\max}]$, for the offline case*

$$r_{u,v} := \mathrm{Cov}\left(Z'_u, Z'_v\right) = \sqrt{\binom{u}{2}\binom{v}{2}} \Big/ \binom{u \vee v}{2}, \quad (7)$$

*where $u \vee v = \max\{u, v\}$, and for the online case,*

$$r'_{u,v} := \mathrm{Cov}(M_u, M_{u+s}) = (1 - \frac{s}{B_0})(1 - \frac{s}{B_0 - 1}), \text{ for } s \geq 0.$$

In the offline setting, the choice of the threshold $b$ involves a tradeoff between two standard performance metrics: (**i**) the significant level (SL), which is the probability that the $M$-statistic exceeds the threshold $b$ under the null hypothesis (i.e., when there is no change-point); and (**ii**) power, which is the probability of the statistic exceeds the threshold under the alternative hypothesis. In the online setting, there are two analogous performance metrics commonly used for analyzing change-point detection procedures [15]: (**i**) the expected value of the stopping time when there is no change, the average run length (ARL); (**ii**) the expected detection delay (EDD), defined to be the expected stopping time in the extreme case where a change occurs immediately at $\tau = 0$. We focus on analyzing SL and ARL of our methods, since they play key roles in setting thresholds. We derive accurate approximations to these quantities as functions of the threshold $b$, so that given a prescribed SL or

ARL, we can solve for the corresponding $b$ analytically. Let $\mathbb{P}^\infty$ and $\mathbb{E}^\infty$ denote, respectively, the probability measure and expectation under the null.

**Theorem 3 (SL in offline case.)** *When $b \to \infty$ and $b/\sqrt{B_{max}} \to c$ for some constant $c$, the significant level of the offline $M$-statistic defined in (3) is given by*

$$\mathbb{P}^\infty \left\{ \max_{B \in \{2,3,\ldots,B_{max}\}} \frac{Z_B}{\sqrt{\text{Var}[Z_B]}} > b \right\} = b^2 e^{-\frac{1}{2}b^2} \cdot \sum_{B=2}^{B_{max}} \frac{(2B-1)}{2\sqrt{2\pi}B(B-1)} \nu \left( b \sqrt{\frac{2B-1}{B(B-1)}} \right) + o(1),$$
(8)

*where the special function $\nu(u) \approx \frac{(2/u)(\Phi(u/2)-0.5)}{(u/2)\Phi(u/2)+\phi(u/2)}$, $\phi$ is the probability density function and $\Phi(x)$ is the cumulative distribution function of the standard normal distribution, respectively.*

The proof of theorem 3 uses a change-of-measure argument, which is based on the likelihood ratio identity (see, e.g., [12, 16]). The likelihood ratio identity relates computing of the tail probability under the null to computing a sum of expectations each under an alternative distribution indexed by a particular parameter value. To illustrate, assume the probability density function (pdf) under the null is $f(u)$. Given a function $g_\omega(x)$, with $\omega$ in some index set $\Omega,$, we may introduce a family of alternative distributions with pdf $f_\omega(u) = e^{\theta g_\omega(u) - \psi_\omega(\theta)} f(u)$, where $\psi_\omega(\theta) := \log \int e^{\theta g_\omega(u)} f(u) du$ is the log moment generating function, and $\theta$ is the parameter that we may assign an arbitrary value. It can be easily verified that $f_\omega(u)$ is a pdf. Using this family of alternative, we may calculate the probability of an event $A$ under the original distribution $f$, by calculating a sum of expectations:

$$\mathbb{P}\{A\} = \mathbb{E}\left[ \frac{\sum_{\omega \in \Omega} e^{\ell_\omega}}{\sum_{s \in \Omega} e^{\ell_s}}; A \right] = \sum_{\omega \in \Omega} \mathbb{E}_\omega[e^{\ell_\omega}; A],$$

where $\mathbb{E}[U; A] := \mathbb{E}[U\mathbb{I}\{A\}]$, the indicator function $\mathbb{I}\{A\}$ is one when event $A$ is true and zero otherwise, $\mathbb{E}_\omega$ is the expectation using pdf $f_\omega(u)$, $\ell_\omega = \log[f(u)/f_\omega(u)] = \theta g_\omega(u) - \psi_\omega(\theta)$, is the log-likelyhood ratio, and we have the freedom to choose a different $\theta$ value for each $f_\omega$.

The basic idea of change-of-measure in our setting is to treat $Z_B' := Z_B/\text{Var}[Z_B]$, as a random field indexed by $B$. Then to characterize SL, we need to study the tail probability of the maximum of this random field. Relate this to the setting above, $Z_B'$ corresponds to $g_\omega(u)$, $B$ corresponds to $\omega$, and $A$ corresponds to the threshold crossing event. To compute the expectations under the alternative measures, we will take a few steps. First, we choose a parameter value $\theta_B$ for each pdf associated with a parameter value $B$, such that $\dot\psi_B(\theta_B) = b$. This is equivalent to setting the mean under each alternative probability to the threshold $b$: $\mathbb{E}_B[Z_B'] = b$ and it allows us to use the local central limit theorem since under the alternative measure boundary cross has much larger probability. Second, we will express the random quantities involved in the expectations, as a functions of the so-called local field terms: $\{\ell_B - \ell_s : s = B, B \pm 1, \ldots\}$, as well as the re-centered log-likelihood ratios: $\tilde\ell_B = \ell_B - b$. We show that they are asymptotically independent as $b \to \infty$ and $b$ grows on the order of $\sqrt{B}$, and this further simplifies our calculation. The last step is to analyze the covariance structure of the random field (Lemma 2 in the following), and approximate it using a Gaussian random field. Note that the terms $Z_u'$ and $Z_v'$ have non-negligible correlation due to our construction: they share the same post-change block $Y^{(B)}$. We then apply the localization theorem (Theorem 5.2 in [16]) to obtain the final result.

**Theorem 4 (ARL in online case.)** *When $b \to \infty$ and $b/\sqrt{B_0} \to c'$ for some constant $c'$, the average run length (ARL) of the stopping time $T$ defined in (5) is given by*

$$\mathbb{E}^\infty[T] = \frac{e^{b^2/2}}{b^2} \cdot \left\{ \frac{(2B_0-1)}{\sqrt{2\pi}B_0(B_0-1)} \cdot \nu \left( b \sqrt{\frac{2(2B_0-1)}{B_0(B_0-1)}} \right) \right\}^{-1} + o(1).$$
(9)

Proof for Theorem 4 is similar to that for Theorem 3, due to the fact that for a given $m > 0$,

$$\mathbb{P}^\infty\{T \le m\} = \mathbb{P}^\infty \left\{ \max_{1 \le t \le m} M_t > b \right\}.$$
(10)

Hence, we also need to study the tail probability of the maximum of a random field $M_t = Z_{B_0,t}/\sqrt{Z_{B_0,t}}$ for a fixed block size $B_0$. A similar change-of-measure approach can be used, except that the covariance structure of $M_t$ in the online case is slightly different from the offline case. This tail probability turns out to be in a form of $\mathbb{P}^\infty\{T \le m\} = m\lambda + o(1)$. Using similar argu-

ments as those in [13, 14], we may see that $T$ is asymptotically exponentially distributed. Hence, $\mathbb{P}^\infty\{T \le m\} - [1 - \exp(-\lambda m)] \to 0$. Consequently $\mathbb{E}^\infty\{T\} \sim \lambda^{-1}$, which leads to (9).

Theorem 4 shows that $\text{ARL} \sim \mathcal{O}(e^{b^2})$ and, hence, $b \sim \mathcal{O}(\sqrt{\log \text{ARL}})$. On the other hand, the EDD is typically on the order of $b/\Delta$ using the Wald's identity [12] (although a more careful analysis should be carried out in the future work), where $\Delta$ is the Kullback-Leibler (KL) divergence between the null and alternative distributions (on a order of a constant). Hence, given a desired ARL (typically on the order of 5000 or 10000), the error made in the estimated threshold will only be translated linearly to EDD. This is a blessing to us and it means typically a reasonably accurate $b$ will cause little performance loss in EDD. Similarly, Theorem 3 shows that $\text{SL} \sim \mathcal{O}(e^{-b^2})$ and a similar argument can be made for the offline case.

## 5 Numerical examples

We test the performance of the $M$-statistic using simulation and real world data. Here we only highlight the main results. More details can be found in Appendix C. In the following examples, we use a Gaussian kernel: $k(Y, Y') = \exp\left(-\|Y - Y'\|^2/2\sigma^2\right)$, where $\sigma > 0$ is the kernel bandwidth and we use the "median trick" [10, 8] to get the bandwidth which is estimated using the background data.

**Accuracy of Lemma 1 for estimating** $\text{Var}[Z_B]$**.** Fig. 5 in the appendix shows the empirical distributions of $Z_B$ when $B = 2$ and $B = 200$, when $N = 5$. In both cases, we generate 10000 random instances, which are computed from data following $\mathcal{N}(0, I)$, $I \in \mathbb{R}^{20 \times 20}$ to represent the null distribution. Moreover, we also plot the Gaussian pdf with sample mean and sample variance, which matches well with the empirical distribution. Note the approximation works better when the block size decreases. (The skewness of the statistic can be corrected; see discussions in Section 7).

**Accuracy of theoretical results for estimating threshold.** For the offline case, we compare the thresholds obtained from numerical simulations, bootstrapping, and using our approximation in Theorem 3, for various SL values $\alpha$. We choose the maximum block size to be $B_{\max} = 20$. In the appendix, Fig. 6(a) demonstrates how a threshold is obtained by simulation, for $\alpha = 0.05$, the threshold $b = 2.88$ corresponds to the 95% quantile of the empirical distribution of the offline $M$-statistic. For a range of $b$ values, Fig. 6(b) compares the empirical SL value $\alpha$ from simulation with that predicted by Theorem 3, and shows that theory is quite accurate for small $\alpha$, which is desirable as we usually care of small $\alpha$'s to obtain thresholds. Table 1 shows that our approximation works quite well to determine thresholds given $\alpha$'s: thresholds obtained by our theory matches quite well with that obtained from Monte Carlo simulation (the null distribution is $\mathcal{N}(0, I)$, $I \in \mathbb{R}^{20 \times 20}$), and even from bootstrapping for a real data scenario. Here, the "bootstrap" thresholds are for a speech signal from the CENSREC-1-C dataset. In this case, the null distribution $P$ is unknown, and we only have 3000 samples speech signals. Thus we generate bootstrap samples to estimate the threshold, as shown in Fig. 7 in the appendix. These $b$'s obtained from theoretical approximations have little performance degradation, and we will discuss how to improve in Section 7.

Table 1: Comparison of thresholds for offline case, determined by simulation, bootstrapping and theory respectively, for various SL value $\alpha$.

| $\alpha$ | $B_{\max} = 10$ | | | $B_{\max} = 20$ | | | $B_{\max} = 50$ | | |
|---|---|---|---|---|---|---|---|---|---|
| | $b$ (sim) | $b$ (boot) | $b$ (the) | $b$ (sim) | $b$ (boot) | $b$ (the) | $b$ (sim) | $b$ (boot) | $b$ (the) |
| 0.20 | 1.78 | 1.77 | 2.00 | 1.97 | 2.29 | 2.25 | 2.21 | 2.47 | 2.48 |
| 0.15 | 2.02 | 2.05 | 2.18 | 2.18 | 2.63 | 2.41 | 2.44 | 2.78 | 2.62 |
| 0.10 | 2.29 | 2.45 | 2.40 | 2.47 | 3.09 | 2.60 | 2.70 | 3.25 | 2.80 |

For the online case, we also compare the thresholds obtained from simulation (using 5000 instances) for various ARL and from Theorem 4, respectively. As predicated by theory, the threshold is consistently accurate for various null distributions (shown in Fig. 3). Also note from Fig. 3 that the precision improves as $B_0$ increases. The null distributions we consider include $\mathcal{N}(0, 1)$, exponential distribution with mean 1, a Erdos-Renyi random graph with 10 nodes and probability of 0.2 of forming random edges, and Laplace distribution.

**Expected detection delays (EDD).** In the online setting, we compare EDD (with the assumption $\tau = 0$) of detecting a change-point when the signal is 20 dimensional and the transition happens

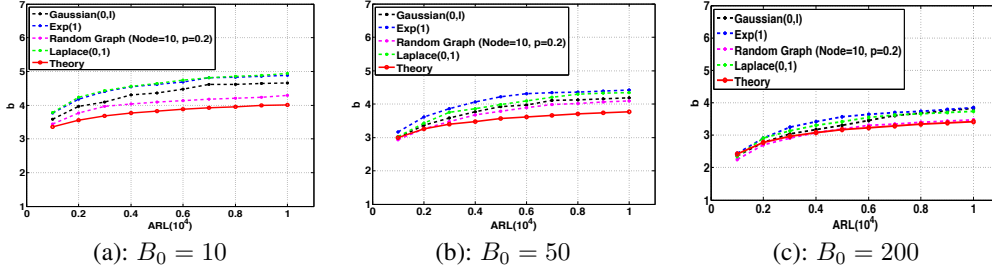

(a): $B_0 = 10$   (b): $B_0 = 50$   (c): $B_0 = 200$

Figure 3: In online case, for a range of ARL values, comparison $b$ obtained from simulation and from Theorem 4 under various null distributions.

from a zero-mean Gaussian $\mathcal{N}(0, I_{20})$ to a non-zero mean Gaussian $\mathcal{N}(\mu, I_{20})$, where the post-change mean vector $\mu$ is element-wise equal to a constant mean shift. In this setting, Fig. 10(a) demonstrates the tradeoff in choosing a block size: when block size is too small the statistical power of the $M$-statistic is weak and hence EDD is large; on the other hand, when block size is too large, although statistical power is good, EDD is also large because the way we update the test block. Therefore, there is an optimal block size for each case. Fig. 10(b) shows the optimal block size decreases as the mean shift increases, as expected.

## 6  Real-data

We test the performance of our $M$-statistics using real data. Our datasets include: (1) CENSREC-1-C: a real-world speech dataset in the Speech Resource Consortium (SRC) corpora provided by National Institute of Informatics (NII)[1]; (2) Human Activity Sensing Consortium (HASC) challenge 2011 data[2]. We compare our $M$-statistic with a state-of-the-art algorithm, the relative density-ratio (RDR) estimate [7] (one limitation of the RDR algorithm, however, is that it is not suitable for high-dimensional data because estimating density ratio in the high-dimensional setting is ill-posed). To achieve reasonable performance for the RDR algorithm, we adjust the bandwidth and the regularization parameter at each time step and, hence, the RDR algorithm is computationally more expensive than the $M$-statistics method. We use the Area Under Curve (AUC) [7] (the larger the better) as a performance metric. Our $M$-statistics have competitive performance compared with the baseline RDR algorithm in the real data testing. Here we report the main results and the details can be found in Appendix D. For the speech data, our goal is to detect the onset of speech signal emergent from the background noise (the background noises are taken from real acoustic signals, such as background noise in highway, airport and subway stations). The overall AUC for the $M$-statistic is **.8014** and for the baseline algorithm is **.7578**. For human activity detection data, we aim at detection the onset of transitioning from one activity to another. Each data consists of human activity information collected by portable three-axis accelerometers. The overall AUC for the $M$-statistic is **.8871** and for the baseline algorithm is **.7161**.

## 7  Discussions

We may be able to improve the precision of the tail probability approximation in theorems 3 and 4 to account for skewness of $Z'_B$. In the change-of-measurement argument, we need to choose parameter values $\theta_B$ such that $\dot{\psi}_B(\theta_B) = b$. Currently, we use a Gaussian assumption $Z'_B \sim \mathcal{N}(0, 1)$ and, hence, $\psi_B(\theta) = \theta^2/2$, and $\theta_B = b$. We may improve the precision if we are able to estimate skewness $\kappa(Z'_B)$ for $Z'_B$. In particular, we can include the skewness in the log moment generating function approximation $\psi_B(\theta) \approx \theta^2/2 + \kappa(Z'_B)\theta^3/6$ when we estimate the change-of-measurement parameter: setting the derivative of this to $b$ and solving a quadratic equation $\kappa(Z'_B)\theta^2/2 + \theta = b$ for $\theta'_B$. This will change the leading exponent term in (8) from $e^{-b^2/2}$ to be $e^{\psi'_B(\theta'_B) - \theta'_B b}$. A similar improvement can be done for the ARL approximation in Theorem 4.

**Acknowledgments**

This research was supported in part by CMMI-1538746 and CCF-1442635 to Y.X.; NSF/NIH BIGDATA 1R01GM108341, ONR N00014-15-1-2340, NSF IIS-1218749, NSF CAREER IIS-1350983 to L.S..

## Footnotes

[1] Available from http://research.nii.ac.jp/src/en/CENSREC-1-C.html

[2] Available from http://hasc.jp/hc2011

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
