[Supplementary Material · nips2015_nonparamatric_changepoint_camera_app.pdf]

# A Recursive update for online $M$-statistic.

There is an efficient way to update the online $M$-statistic. Recall that we generate the statistic as follows. When time goes from $t$ to $t+1$, we add the new sample into the post-change block, remove the oldest sample and move it to the reference pool. We will update the reference blocks similarly: draw a batch of $N$ distinct samples from the pool, add them to each of the reference block, and purge the oldest sample from each reference block. Hence, we only need to compare a few number of MMD statistic, due to the new samples added to the reference block and the new sample at time $t+1$. These calculations are illustrated in Fig. 4. At time $t$, for every paired blocks $X_i^{(B_0,t)}$ and $Y^{(B_0,t)}$, we compute the Gram matrix: for $N$ background blocks and one testing blocks, we have $N$ such Gram matrices. We partition the Gram matrix by four windows (in red, black and blue, as shown on the left). To get $\text{MMD}^2(X_i^{(B_0,t)}, Y^{(B_0,t)})$, we compute the shaded elements and take an average within each the window. The diagonal entries in each window are removed since we are using the unbiased expression for MMD for two-sample test when the two sample sizes are equal. At time $t+1$, we update $X_i^{(B_0,t)}$ and $Y^{(B_0,t)}$ with a new data and purge the oldest data. Consequently, for the Gram matrix, we move the colored window as shown on the right figure, compute the elements within the new windows, and take an average. Note that for each window, there are $(B_0-1)(B_0-2)$ elements shared in common with the previous windows, thus we only need to compute the right-most column and the bottom row. This way we recursively update the kernel matrices and through which compute the statistic.

Figure 4: Update the Gram matrix used in calculating the online $M$-statistics.

# B Proofs

We start with proving Lemma 5 and Lemma 6, which are useful in proving Lemma 1 and Lemma 2.

**Lemma 5 (Variance of MMD, under the null.)** *Under null hypothesis,*

$$\mathrm{Var}\left[\mathrm{MMD}^2(X_i^{(B)}, Y^{(B)})\right] = \binom{B}{2}^{-1} \mathbb{E}[h^2(x, x', y, y')], \quad i = 1, \ldots, N. \tag{11}$$

**Proof** [Proof of Lemma 5] For notational simplicity, we drop the superscript $B$. Furthermore, under the null hypothesis all data follow the same distribution, we can represent $X_{i,l}$ and $X_{i,j}$ as $x$ and $x'$, and $Y_l$ and $Y_j$ as $y$ and $y'$, respectively. For any $i = 1, 2, \ldots, n$, by definition of U-statistic, we have

$$\mathrm{Var}\left[\mathrm{MMD}^2(X_i, Y)\right] = \mathrm{Var}\left[\binom{B}{2}^{-1} \sum_{l<j} h(X_{i,l}, X_{i,j}, Y_l, Y_j)\right]$$

$$= \binom{B}{2}^{-2}\left[\binom{B}{2}\binom{2}{1}\binom{B-2}{2-1}\mathrm{Var}\left[\mathbb{E}_{x_i y}[h(x, x', y, y')]\right] + \binom{B}{2}\binom{2}{2}\binom{B-2}{2-2}\mathrm{Var}\left[h(x, x', y, y')\right]\right]. \tag{12}$$

Under null distribution, $\mathbb{E}_{x_i y}[h(x, x', y, y')] = 0$. Thus, $\mathrm{Var}\left[\mathbb{E}_{x_i y}[h(x, x', y, y')]\right] = 0$, and

$$\mathrm{Var}\left[h(x, x', y, y')\right] = \mathbb{E}[h^2(x, x', y, y')] - \mathbb{E}[h(x, x', y, y')]^2 = \mathbb{E}[h^2(x, x', y, y')].$$

Substitute these results in (12), we obtain the desired result (11).

■

**Lemma 6 (Covariance of MMD, under the null, same block size.)** *For $s \neq 0$, under null hypothesis*

$$\mathrm{Cov}\left[\mathrm{MMD}^2(X_i^{(B)}, Y^{(B)}), \mathrm{MMD}^2(X_{i+s}^{(B)}, Y^{(B)})\right] = \binom{B}{2}^{-1}\mathrm{Cov}\left[h(x_i, x_i', y, y'), h(x_{i+s}, x_{i+s}', y, y')\right].$$

**Proof** [Proof of Lemma 6] For notational simplicity, we drop the superscript $B$. For $i = 1, 2, \ldots, N$, and $s = (1-i), (2-i), \ldots, (N-i), s \neq 0$,

$$\mathrm{Cov}\left[\mathrm{MMD}^2(X_i, Y), \mathrm{MMD}^2(X_{i+s}, Y)\right]$$

$$= \mathrm{Cov}\left[\binom{B}{2}^{-1}\sum_{l<j}h(X_{i,l}, X_{i,j}, Y_l, Y_j), \binom{B}{2}^{-1}\sum_{p<q}h(X_{i+s,p}, X_{i+s,q}, Y_p, Y_q)\right]$$

$$= \binom{B}{2}^{-2}\binom{B}{2}\binom{2}{1}\binom{B-2}{2-1}\mathrm{Cov}\left[h(X_{i,l}, X_{i,j}, y, Y_j), h(X_{i+s,p}, X_{i+s,q}, Y_p, Y_q)\right]$$

$$+ \binom{B}{2}^{-2}\binom{B}{2}\binom{2}{2}\binom{B-2}{2-2}\mathrm{Cov}\left[h(X_{i,l}, X_{i,j}, y, y'), h(X_{i+s,p}, X_{i+s,q}, y, y')\right].$$

Under null distribution,

$$\mathrm{Cov}\left[h(X_{i,l}, X_{i,j}, y, Y_j), h(X_{i+s,p}, X_{i+s,q}, y, Y_q)\right]$$

$$= \int \mathbb{P}[X_{i,l}, X_{i,j}, y, Y_j, X_{i+s,p}, X_{i+s,q}, y, Y_q]h(X_{i,l}, X_{i,j}, y, Y_j)h(X_{i+s,p}, X_{i+s,q}, y, Y_q)$$

$$= \int \mathbb{P}[X_{i,l}, y]\mathbb{P}[X_{i+s,p}, y]\int \mathbb{P}[X_{i,j}, Y_j]h(X_{i,l}, X_{i,j}, y, Y_j)\int \mathbb{P}[X_{i+s,q}, Y_q]h(X_{i+s,p}, X_{i+s,q}, y, Y_q) = 0.$$

Finally, we have:

$$\mathrm{Cov}\left[\mathrm{MMD}^2(X_i, Y), \mathrm{MMD}^2(X_{i+s}, Y)\right] = \binom{B}{2}^{-1}\mathrm{Cov}\left[h(X_{i,l}, X_{i,j}, y, y'), h(X_{i+s,p}, X_{i+s,q}, y, y')\right].$$

Under null hypothesis, $X_{i,l}$, $X_{i,j}$, $X_{i+s,p}$, and $X_{i+s,q}$ are independent and they follow the same null distribution, so we may replace them with $x$, $x'$, $x''$, $x'''$ respectively. Finally

$$\text{Cov}\left[\text{MMD}^2(X_i, Y; B), \text{MMD}^2(X_{i+s}, Y; B)\right] = \binom{B}{2}^{-1} \text{Cov}\left[h(x, x', y, y'), h(x'', x''', y, y')\right].$$

∎

**Proof** [Proof for Lemma 1] For notational simplicity, we drop the superscript $B$. Using results in Lemma 5 and Lemma 6, we have

$$\text{Var}[Z_B] = \text{Var}\left[\frac{1}{N}\sum_{i=1}^N \text{MMD}^2(X_i, Y)\right]$$

$$= \frac{1}{N^2}\left[N\text{Var}[\text{MMD}^2(X_i, Y)] + \sum_{i\neq j}\text{Cov}\left[\text{MMD}^2(X_i, Y; B), \text{MMD}^2(X_j, Y)\right]\right]$$

$$= \frac{1}{N}\binom{B}{2}^{-1}\mathbb{E}[h^2(x_i, x_i', y, y')] + \frac{1}{N^2}\sum_{i\neq j}\binom{B}{2}^{-1}\text{Cov}\left[h(x_i, x_i', y, y'), h(x_j, x_j', y, y')\right]$$

$$= \binom{B}{2}^{-1}\left[\frac{1}{N}\mathbb{E}[h^2(x, x', y, y')] + \frac{N-1}{N}\text{Cov}\left[h(x, x', y, y'), h(x'', x''', y, y')\right]\right].$$

∎

**Proof** [Proof of Lemma 2] For the offline case, we have that the correlation

$$r_{B,B+v} = \frac{1}{\sqrt{\text{Var}[Z_B]}}\frac{1}{\sqrt{\text{Var}[Z_{B+v}]}}\text{Cov}\left[Z_B, Z_{B+v}\right],$$

where

$$\text{Cov}\left(Z_B, Z_{B+v}\right) = \text{Cov}\left[\frac{1}{N}\sum_{i=1}^N \text{MMD}^2(X_i^{(B)}, Y^{(B)}), \frac{1}{N}\sum_{j=1}^n \text{MMD}^2(X_j^{(B+v)}, Y^{(B+v)})\right]$$

$$= \frac{1}{N}\text{Cov}\left[\text{MMD}^2(X_i^{(B)}, Y^{(B)}), \text{MMD}^2(X_i^{(B+v)}, Y^{(B+v)})\right]$$

$$+ \frac{1}{N^2}\sum_{i\neq j}\text{Cov}\left[\text{MMD}^2(X_i^{(B)}, Y^{(B)}), \text{MMD}^2(X_j^{(B+v)}, Y^{(B+v)})\right].$$

Using results from Lemma 5 and Lemma 6, we have:

$$\text{Cov}\left(Z_B, Z_{B+v}\right) = \frac{1}{N}\binom{B\vee(B+v)}{2}^{-1}\mathbb{E}[h^2(x, x', y, y')]$$

$$+ \frac{N-1}{N}\binom{B\vee(B+v)}{2}^{-1}\text{Cov}\left[h(x, x', y, y'), h(x'', x''', y, y')\right]$$

$$= \binom{B\vee(B+v)}{2}^{-1}\left[\frac{1}{N}\mathbb{E}[h^2(x, x', y, y')] + \frac{N-1}{N}\text{Cov}\left[h(x, x', y, y'), h(x'', x''', y, y')\right]\right].$$

Finally, plugging in the expressions for $\text{Var}[Z_B]$ and $\text{Var}[B+v]$, we have (7) for the offline case.

For the online case we need to analyze $r' = \text{Cov}(M_t, M_{t+s})$. Without loss of generality, assume $s > 0$. We may use the covariance result above for a fixed block size $B_0$ to obtain

$$\text{Cov}\left(\text{MMD}^2(X_i^{(B_0,t)}, Y^{(B_0,t)}), \text{MMD}^2(X_i^{(B_0,t+s)}, Y^{(B_0,t+s)})\right)$$

$$= \binom{B}{2}^{-2}\binom{B-s}{2}\text{Var}[h(x, x', y, y')], \tag{13}$$

and

$$\mathrm{Cov}\left(\mathrm{MMD}^2(X_i^{(B_0,t)}, Y^{(B_0,t)}), \mathrm{MMD}^2(X_j^{(B_0,t+s)}, Y^{(B_0,t+s)})\right)$$

$$= \binom{B}{2}^{-2}\binom{B-s}{2}\mathrm{Cov}(h(x,x',y,y'), h(x'',x''',y,y')). \tag{14}$$

Thus,

$$\mathrm{Cov}\left(Z_{B_0,t}, Z_{B_0,k+s}\right)$$

$$= \mathrm{Cov}\left(\frac{1}{N}\sum_{i=1}^N \mathrm{MMD}^2(X_i^{(B_0,t)}, Y^{(B_0,t)}), \frac{1}{N}\sum_{j=1}^N \mathrm{MMD}^2(X_j^{(B_0,t+s)}, Y^{(B_0,t+s)})\right)$$

$$= \binom{B_0}{2}^{-2}\binom{B_0-s}{2}\left[\frac{1}{N}\mathrm{Var}(h(x,x',y,y')) + \frac{N-1}{N}\mathrm{Cov}(h(x,x',y,y'), h(x'',x''',y,y'))\right]$$
$$\tag{15}$$

We have:

$$r'_{t,t+s} = \frac{\binom{B_0-s}{2}}{\binom{B_0}{2}} = \left(1 - \frac{s}{B_0}\right)\left(1 - \frac{s}{B_0-1}\right). \tag{16}$$

∎

**Lemma 7 (Covariance of MMD, under the null, different block sizes, same block index.)** *For blocks with the same index $i$ but with distinct block sizes, under the null hypothesis we have*

$$\mathrm{Cov}\left[\mathrm{MMD}^2(X_i, Y; B), \mathrm{MMD}^2(X_i, Y; B+v)\right] = \binom{B \vee (B+v)}{2}^{-1}\mathbb{E}[h^2(x,x',y,y')] \tag{17}$$

**Proof** [Proof of Lemma 7] Note that

$$\mathrm{Cov}\left[\mathrm{MMD}^2(X_i^{(B)}, Y^{(B)}), \mathrm{MMD}^2(X_i^{(B+v)}, Y^{(B+v)})\right]$$

$$= \mathrm{Cov}\left[\binom{B}{2}^{-1}\sum_{l<j}^B h(X_{i,l}, X_{i,j}, Y_l, Y_j), \binom{B+v}{2}^{-1}\sum_{p<q}^{B+v} h(X_{i,p}, X_{i,q}, Y_p, Y_q)\right]$$

$$= \binom{B}{2}^{-1}\binom{B+v}{2}^{-1}\mathrm{Cov}\left[\sum_{l<j}^B h(X_{i,l}, X_{i,j}, Y_l, Y_j), \sum_{p<q}^{B+v} h(X_{i,p}, X_{i,q}, Y_p, Y_q)\right]$$

$$= \binom{B}{2}^{-1}\binom{B+v}{2}^{-1}\binom{B \wedge (B+v)}{2}\mathrm{Var}[h(x,x',y,y')]$$

$$= \binom{B \vee (B+v)}{2}^{-1}\mathbb{E}[h^2(x,x',y,y')],$$

where the second last equality is due to a similar argument as before to drop block indices as they are *i.i.d.* under the null. ∎

**Lemma 8 (Covariance of MMD, under the null, different block sizes and different block indices.)** *Under the null we have*

$$\mathrm{Cov}\left[\mathrm{MMD}^2(X_i^{(B)}, Y^{(B)}), \mathrm{MMD}^2(X_{i+s}^{(B+v)}, Y^{(B+v)})\right] = \binom{B \vee (B+v)}{2}^{-1}\mathrm{Cov}\left[h(x,x',y,y'), h(x'',x''',y,y')\right].$$

**Proof** [Proof of Lemma 8] Note that

$$\text{Cov}\left[\text{MMD}^2(X_i^{(B)}, Y^{(B)}), \text{MMD}^2(X_{i+s}^{(B+v)}, Y^{(B+v)})\right]$$

$$= \text{Cov}\left[\binom{B}{2}^{-1}\sum_{l<j} h(X_{i,l}^{(B)}, X_{i,j}^{(B)}, Y_l^{(B)}, Y_j^{(B)}), \binom{B+v}{2}^{-1}\sum_{p<q}^{B+v} h(X_{i+s,p}^{(B+v)}, X_{i+s,q}^{(B+v)}, Y_p^{(B+v)}, Y_q^{(B+v)})\right]$$

$$= \binom{B}{2}^{-1}\binom{B+v}{2}^{-1}\text{Cov}\left[\sum_{l<j}^{B} h(X_{i,l}^{(B)}, X_{i,j}^{(B)}, Y_l^{(B)}, Y_j^{(B)}), \sum_{p<q}^{B+v} h(X_{i+s,p}^{(B+v)}, X_{i+s,q}^{(B+v)}, Y_p^{(B+v)}, Y_q^{(B+v)})\right]$$

$$= \binom{B}{2}^{-1}\binom{B+v}{2}^{-1}\binom{B\wedge(B+v)}{2}\text{Cov}\left[h(x, x', y, y'), h(x'', x''', y, y')\right]$$

$$= \binom{B\vee(B+v)}{2}^{-1}\text{Cov}\left[h(x, x', y, y'), h(x'', x''', y, y')\right],$$

where the second last equality is due to a similar argument as before to drop block indices as they are *i.i.d.* under the null.

∎

**Proof** [Proof of Theorem 3.] Define $Z_B' = Z_B/\sqrt{\text{Var}[Z_B]}$. We would like to study $\mathbb{P}^\infty\left\{\max_{B\in[2,M]} Z_B' > b\right\}$ under null hypothesis. Recall that $\xi_B$ is set to the solution to $\dot{\psi}_B(\theta) = b$ and $\psi_B(\theta) = \log\mathbb{E}[e^{\theta Z_B'}]$ is the log moment generating function. Under null hypothesis, we may approximate the distribution $Z_B' \sim \mathcal{N}(0,1)$. Hence, $\psi_B(\theta_B) = \theta^2/2$, and the solution $\theta_B$ to $\dot{\psi}(\theta) = b$ becomes

$$\theta_B = b, \quad \text{and} \quad \psi_B(\theta_B) = b^2/2.$$

In the following we will use the "likelihood ratio identity" trick, which computes a probability of an event formulated in some distribution by reformulating it as an expectation in the context of an alternative distribution [12, 16]. We use the notation $\mathbb{E}_B[U; A]$ to indicate that the expectation involves the product between the random variable $U$ and the indicator of the event $A$. Associate with each $B$, $B \in [2, M]$ a log-likelhood ratio of the form

$$\ell_B = \theta_B Z_B' - \psi_B(\theta_B) = bZ_B' - b^2/2. \tag{18}$$

With the aid of such log-likelihood ratios we may produce the likelihood ratio identity:

$$\mathbb{P}^\infty\left\{\max_{B\in[2,B_{\max}]} Z_B' > b\right\} = \mathbb{E}\left[\underbrace{\frac{\sum_{B=2}^{B_{\max}} e^{\ell_B}}{\sum_{s=2}^{B_{\max}} e^{\ell_s}}}_{=1}; \max_{B\in[2,B_{\max}]} Z_B' > b\right]$$

$$= \sum_{B=2}^{B_{\max}}\mathbb{E}\left[\frac{e^{\ell_B}}{\sum_s e^{\ell_s}}; \max_{B\in[2,B_{\max}]} Z_B' > b\right] = \sum_{B=2}^{B_{\max}}\mathbb{E}_B\left[\frac{1}{\sum_s e^{\ell_s}}; \max_{B\in[2,B_{\max}]} Z_B' > b\right], \tag{19}$$

where $\mathbb{P}_B$ is the alternative distribution that is associated with the likelihood ratio $\ell_B$, and

$$\mathbb{E}_B[U] = \mathbb{E}[Ue^{\theta Z_B' - \phi(\theta)}].$$

A local random field is produced by the consideration of difference between the log-likelihood ratio at $B$ and the log-likelihood ratios at other parameter values for the block size. Using (18), the components of the local field are:

$$\ell_s - \ell_B = b(Z_s' - Z_B'). \tag{20}$$

Our approximation will depend on the summation and maximization statistics of the local field:

$$M_B = \max_{B\in[2,B_{\max}]} e^{\ell_s - \ell_B}, \quad \text{and} \quad S_B = \max_{B\in[2,B_{\max}]} e^{\ell_s - \ell_B}.$$

Also introduce the re-centered log-likelihood ratio:

$$\tilde{\ell}_B := \theta_B(Z_B' - \dot{\psi}(\theta_B)) = b(Z_B' - b),$$

By introducing and subtracting or dividing terms in (19), we may write it in a form that is convenient to apply Theorem 5.2 in [16]:

$$\sum_{B=2}^{B_{\max}} e^{\psi_B(\theta_B)-\theta_B b} \mathbb{E}_B \left[ \frac{e^{\theta_B \max_{s\in[2,B]}\left\{Z'_s-Z'_B\right\}} e^{-\theta_B\left[Z'_B-b+\max_{s\in[2,B_{\max}]}\left\{Z'_s-Z'_B\right\}\right]}}{\sum_{s\in[2,B_{\max}]} e^{\theta_B Z'_s-\psi_B(\theta_B)}}; \right.$$

$$\left. Z'_B-b+\max_{s\in[2,B_{\max}]}\{Z'_s-Z'_B\}\geq 0\right] \tag{21}$$

$$= e^{-b^2/2}\sum_{B=2}^{B_{\max}}\mathbb{E}_B\left[\frac{M_B}{S_B}e^{-[\tilde{\ell}_B+\log M_B]};\tilde{\ell}_B+\log M_B\geq 0\right].$$

In order to apply the localization theorem (Theorem 5.2 in [16]) we need to identify the local limit distribution of $\tilde{\ell}_B$ and of the local field $\{\ell_s-\ell_B : s\in[2,B_{\max}]\}$ and prove asymptotic independence between them. The analysis of the limiting distributions should be done under the alternative distribution $\mathbb{P}_B$. Under the alternative distribution $\mathbb{P}_B$, we get that $\mathbb{E}_B[\tilde{\ell}_B]=0$, since $\mathbb{E}_B[\ell_B]=b$, and the variance is $\mathrm{Var}_B(\tilde{\ell}_B)=b^2\mathrm{Var}_B(\ell_B)=b^2\ddot{\psi}_B(\theta_B)=b^2$, since $\psi_B(\theta)=\theta^2/2$. On the other hand, using a decomposition technique similar to that is used for the proof of Lemma 9, the covariance between the local field $\{\ell_s-\ell_B\}$ and the re-centered log-likelihood ratio $\tilde{\ell}_B$ is given by

$$\mathrm{Cov}(\ell_s-\ell_B,\tilde{\ell}_B)=\mathbb{E}_B[b(Z_s-Z_B)\cdot b(Z_B-b)]=-b^2(1-r_{s,B})\mathbb{E}_B[Z_B(Z_B-b)]$$
$$=-b^2(1-r_{s,B})\approx -b^2\frac{1}{2}\frac{2(B-1)}{B(B-1)}|B-s|. \tag{22}$$

Hence, the asymptotic independence between the local field and the re-centered log-likelihood ratio follows from the fact that, when $b\to\infty$ and $b/\sqrt{B}\to c$ for some constant $c$, if $|B-s|$ is small, the covariance between $\ell_s-\ell_B$ and $\tilde{\ell}_B$ is on the order of a constant. However, the standard deviation of $\tilde{\ell}_B$ diverges to infinity proportional to $b$. Consequently, the correlation between the global term and local fields tends to 0.

We will approximate the limit joint distribution of the local field and the global term is Gaussian. Computation of the expectation and covariance structure are sufficient for obtain the final approximation. Lemma 9 shows that the asymptotic distribution of $\{\ell_s-\ell_B\}$, for $s=B+j$ and $|j|$ not too large, is a two-sided Gaussian random walk with a negative drift. The variance of an increment of this random walk is $\mu^2$.

Using the localization theorem (Theorem 5.2 in [16]), since the local field and the re-centered log-likelihood ratio are asymptotically independent when $b\to\infty$, we have

$$\mathbb{E}_B\left[\frac{M_B}{S_B}e^{-[\tilde{\ell}_B+\log M_B]};\tilde{\ell}_B+\log M_B\geq 0\right]\approx\frac{\mu^2}{2}\nu(\mu)\frac{1}{\sqrt{2\pi\ddot{\psi}(\theta_B)}}=\frac{\mu^2\nu(\mu)}{2\sqrt{2\pi}}. \tag{23}$$

Finally, combine the results above we obtain (8). ∎

**Lemma 9 (Offline, analysis of mean and variance of local field. )** *The mean and variance of the local field $\{\ell_{B+v}-\ell_B\}$, for $v=0,\pm 1,\pm 2,\ldots$, are related by*

$$\mathbb{E}_B[\ell_{B+v}-\ell_B]=-\frac{1}{2}\mathrm{Var}_B[\ell_{B+v}-\ell_B]. \tag{24}$$

*Moreover, given $\mu=b\sqrt{\frac{2B-1}{B(B-1)}}$,*

$$\mathbb{E}_B[\ell_{B+v}-\ell_B]\approx -\frac{\mu^2}{2}|v|,\quad \mathrm{Var}_B[\ell_{B+v}-\ell_B]\approx\mu^2|v|. \tag{25}$$

**Proof** [Proof of Lemma 9] From the definition of the local field (9), we have that for $s=B+v$:

$$\mathbb{E}_B\left[\ell_{B+v}-\ell_B\right]=\mathbb{E}_B\left[b(Z'_{B+v}-Z'_B)\right]=\mathbb{E}\left[b(Z'_{B+v}-Z'_B)e^{bZ'_B-b^2/2}\right]$$
$$=\mathbb{E}\left[\left(-b(1-r_{B+v,B})Z'_B+b\sqrt{1-r^2_{B+v,B}}W\right)e^{bZ'_B-b^2/2}\right]. \tag{26}$$

The above representation results from the regression of $Z'_{B+v}$ on $Z'_B$:

$$Z'_{B+v} = r_{B+v,B}Z'_B + \sqrt{1 - r^2_{B+v,B}}W,$$

with $W$ being the standardized residual of the regression, and $r = \text{Cov}\left(Z'_B, Z'_{B+v}\right)$. Since $W$ is zero-mean and independent of $Z'_B$, (26) becomes

$$\mathbb{E}_B[\ell_{B+v} - \ell_B] = -b(1-r)\mathbb{E}\left[Z'_B e^{bZ'_B - b^2/2}\right] = -b^2(1-r), \tag{27}$$

and the last equality follows from the Gaussianity $Z'_B \sim \mathcal{N}(0,1)$:

$$\mathbb{E}\left[Z'_B e^{bZ'_B - \frac{1}{2}b^2}\right] = \frac{1}{\sqrt{2\pi}}\int u e^{bu - b^2/2} \cdot e^{-u^2/2} du = \frac{1}{\sqrt{2\pi}}\int u e^{-\frac{(u-b)^2}{2}} = b. \tag{28}$$

Similarly, we can compute the variance of the local field under the transformed measure

$$\begin{aligned}
\text{Var}_B[\ell_{B+v} - \ell_B] &= \text{Var}_B\left[b(Z'_{B+v} - Z'_B)\right] = \text{Var}_B\left[brZ'_B + b\left[\sqrt{1 - r^2}W\right] - Z'_B\right] \\
&= \text{Var}_B\left[b\sqrt{1 - r^2}W\right] + \text{Var}_B\left[b(r-1)Z'_B\right] \\
&= \mathbb{E}_B[b^2(1-r^2)W^2] - \mathbb{E}_B[b\sqrt{1-r^2}W]^2 + \mathbb{E}_B\left[b^2(r-1)^2(Z'_B)^2\right] - \mathbb{E}_B\left[b(r-1)Z'_B\right]^2 \\
&= b^2(1-r^2) + b^2(r-1)^2 = 2b^2(1-r).
\end{aligned}$$

Hence, we have the desired result (24).

Next, using results from Lemma 2, we have that

$$r_{B,B+v} = \text{Cov}\left[Z'_B, Z'_{B+v}\right] = \sqrt{\binom{B}{2}\binom{B+v}{2}}\bigg/\binom{B \vee (B+v)}{2}. \tag{29}$$

We will linearize $r$ in terms of small increment $v$. For $v > 0$, using Taylor's expansion $(1+u)^{-1} = 1 - x + o(u)$:

$$r_{B,B+v} = \sqrt{\frac{B(B-1)}{(B+v)(B+v-1)}} = \sqrt{\left(1 + \frac{v}{B}\right)^{-1}\left(1 + \frac{v}{B-1}\right)^{-1}} \approx \sqrt{\left(1 - \frac{v}{B}\right)\left(1 - \frac{v}{B-1}\right)}, \tag{30}$$

and for $v < 0$,

$$r_{B,B+v} = \sqrt{\frac{(B+v)(B+v-1)}{B(B-1)}} = \sqrt{\left(1 + \frac{v}{B}\right)\left(1 + \frac{v}{B-1}\right)}. \tag{31}$$

Combine these two cases, we have

$$r_{B,B+v} = \sqrt{\left(1 - \frac{|v|}{B}\right)\left(1 - \frac{|v|}{B-1}\right)} + o(|v|) = 1 - \frac{1}{2}\frac{2B-1}{B(B-1)}|v| + o(|v|). \tag{32}$$

Substitute this in (27), we have that

$$\mathbb{E}_B[\ell_{B+v} - \ell_B] = -\frac{b^2}{2}\frac{2B-1}{B(B-1)}|v| + o(|v|) = -\frac{\mu^2}{2}|v| + o(|v|). \tag{33}$$

$\blacksquare$

**Lemma 10 (Tail of statistics under the null)** *When* $b \to \infty$,

$$\mathbb{P}^\infty\{T < m\} = \mathbb{P}^\infty\left\{\max_{0 < t < m}\frac{Z_{B_0,t}}{\sqrt{\text{Var}\left[Z_{B_0,t}\right]}} > b\right\} = m e^{-\frac{1}{2}b^2} \cdot \frac{b^2(2B-1)\nu\left(b\sqrt{\frac{2(2B-1)}{B(B-1)}}\right)}{B(B-1)\sqrt{2\pi}} + o(1). \tag{34}$$

**Proof** [Proof for Theorem 4] Let $Z'_t := Z_{B_0,t}/\sqrt{\mathrm{Var}[Z_{B_0,t}]}$. We start with finding the tail probability of the online detection statistic. Note that

$$\mathbb{P}^{\infty}(T < m) = \mathbb{P}^{\infty}\left(\max_{1 \le t \le m} M_t > b\right) = \mathbb{P}^{\infty}\left\{\max_{1 \le t \le m} \frac{Z_{B_0,t}}{\sqrt{\mathrm{Var}[Z_{B_0,t}]}} > b\right\} \qquad (35)$$

Since the block size is fixed to be $B_0$, using Lemma 1, we have that

$$\mathrm{Var}(Z'_t) = \mathrm{Var}(Z'_{t+s}) = \binom{B_0}{2}^{-1}\left[\frac{1}{N}\mathrm{Var}(h(x,x',y,y')) + \frac{N-1}{N}\mathrm{Cov}(h(x,x',y,y'),h(x'',x''',y,y'))\right] \qquad (36)$$

Similar to previous analysis, we analyze the local field $\{\ell'_{t+s} - \ell'_t\}$ where

$$\ell'_t := bZ'_t - b^2/2, \quad \ell'_{t+s} := bZ'_{t+s} - b^2/2. \qquad (37)$$

Use a similar change-of-measure argument, for the sequential problem, we have that (35) can be written as

$$e^{-\frac{1}{2}b^2}\sum_{t=1}^{m}\mathbb{E}_t\left(\frac{\mathcal{M}_t}{\mathcal{S}_t}e^{-[\tilde{\ell}'_t + m_t]}; \tilde{\ell}'_t + m_t \ge 0\right), \qquad (38)$$

where the maximum and the sum of the local fields are

$$\mathcal{M}_t = \max_{t \in [1,m]} e^{\ell_m - \ell_t}, \quad \mathcal{S}_t = \sum_{t \in [1,m]} e^{\ell_m - \ell_t} \qquad (39)$$

and the re-centered log-likelihood ratio is given by

$$\tilde{\ell}_t = \ell_t - b, \quad m_t = \log \mathcal{M}_t. \qquad (40)$$

From Lemma 2, $r'_{t,t+s} \approx 1 - \frac{2B_0-1}{B_0(B_0-1)}s$. With similar analysis as for the offline case, we can also show that the mean and the variance of the local field terms are

$$\mathbb{E}_t\{\ell_{t+s} - \ell_t\} = -b^2(1 - r'_{t+s,t}) = -b^2\underbrace{\frac{2B-1}{B(B-1)}}_{\mu^2/2}|s|, \quad \mathrm{Var}_t\{\ell_{t+s} - \ell_t\} = b^2\underbrace{\frac{2(2B-1)}{B(B-1)}}_{\mu^2}|s| \quad (41)$$

And similar, we may show that the local field terms and the re-centered log-likelihood ratio are asymptotically independent. Then using the localization theorem (Theorem 5.2 in [16]), we can write (38) as

$$\mathbb{P}^{\infty}\{T \le m\} \approx \frac{1}{\sqrt{2\pi}}e^{-\frac{1}{2}b^2}\sum_{t=1}^{m}\frac{b^2(2B-1)}{B(B-1)} \cdot \nu\left(b\sqrt{\frac{2(2B-1)}{B(B-1)}}\right)$$

$$= m \cdot \frac{e^{-\frac{1}{2}b^2}}{\sqrt{2\pi}}\frac{b^2(2B-1)}{B(B-1)} \cdot \nu\left(b\sqrt{\frac{2(2B-1)}{B(B-1)}}\right), \qquad (42)$$

where the last equation is due to the fact that the terms inside the sum are constants that are independent of $t$. Using similar arguments as those in [13, 14], we may see that $T$ is asymptotically exponentially distributed and is uniformly integrable. Hence, if $\lambda$ denotes the factor multiplying $m$ on the right-hand side of (42), then for still larger $m$, in the range where $m\lambda$ is bounded away from 0 and $\infty$, $\mathbb{P}^{\infty}\{T \le m\} - [1 - \exp(-\lambda m)] \to 0$. Consequently $\mathbb{E}^{\infty}\{T\} \sim \lambda^{-1}$, which is equivalent to (9).

∎

## C  Details for numerical examples.

**Examples $M$-statistics.** Figs. 2(a)-(c) are from the simulated data, where data before the change-point are drawn *i.i.d.*from $\mathcal{N}(0,1)$ and after the change-point at $\tau = 250$ are drawn *i.i.d.*from Laplace$(0,1)$. The lower panels show the offline and online $M$-statistics in different settings. Note that in this case, the mean and variance before and after the change-point are identical, and hence conventional statistics based on mean and variance of the data cannot detect the change; however, the $M$-statistic can detect the change-point in both the offline and the online setting. Also, the theoretical threshold obtained from Theorem 3 and Theorem 4 both effectively detect the change-point. For Fig. 2(d), we select one segment of seismic data, where there is an evident change-point and our online $M$-statistic can cross the theoretical threshold we set and detect the event quickly. Note that the $M$-statistic will be affected by the kernel bandwidth, and choosing the bandwidth using the median trick works for this case.

**Accuracy of Lemma 1 for estimating** $\mathrm{Var}[Z_B]$**.** We use Monte Carlo simulations to form 10000 instances of $Z_B$ using data follow the null distribution. Then we find the sample variance of $Z_B$: when $B = 2$, the sample variance is 0.0073, and when $B = 200$, the sample variance is $3.5220 \times 10^{-7}$. Figs. 6(a) and (b) demonstrate the histograms of $Z_B$ in these two cases and the Gaussian pdf's with the sample mean and sample variance. Figs. 6(c) and (d) show the Q-Q plot for the two cases and illustrate that the Gaussian assumption is reasonable, although there is certain skewness in the statistic.

Note that using direct simulation to obtain sample variances of $Z_B$ requires huge amount of data. For instance, to generate 10000 instances, we need a total of $10000(N + 1)B$ samples from the reference data. On the other hand, estimate variance using Lemma 1 requires much smaller number of samples, as we only need to evaluate $\mathbb{E}(h^2(x, x', y, y'))$ and $\mathrm{Cov}[h(x, x', y, y'), h(x''.x''', y, y')]$. Figs. 6(e) and (f) show the percentage difference between theoretical estimate by Lemma 1, relative those obtained from the complete simulation. It can be seen that the error decreases with more reference data and the estimate is reasonably accurate with moderate amount of reference data.

(a): $B = 2$, $N = 5$, empirical distribution    (b): $B = 200$, $N = 5$, empirical distribution

(c): $B = 2$, $N = 5$, Q-Q plot    (d): $B = 200$, $N = 5$, Q-Q plot

(e): $B = 2$, % error for estimated variance    (f): $B = 200$, % error for estimated variance

Figure 5: Accuracy of Lemma 1 in estimating the variance of $Z_B$ when $B = 2$ and $B = 200$.

**Accuracy of Theorem 3 for offline case.** We compare the approximated tail probability from Theorem 3 with that obtained from the empirical distribution with 5000 instances of $M$-statistics. Assume the data is 20-dimensional and the null distribution is $\mathcal{N}(0, I_{20})$ and set $B_{\max} = 20$. Fig.6 (a) demonstrates how to determine the threshold from empirical distribution and Fig.6(b) demonstrates that the approximation works well especially for small $\alpha$ values.

Furthermore, we also show the thresholds for real speech signals in the CENSREC-1-C dataset, as shown in Fig. 7. In this case, the reference distribution $P$ is the unknown distribution of real speech signal, and we only have a limited number of speech signals and we generate 10000 bootstrap samples to estimate the empirical distribution. Note that in this harder case, the theoretical threshold matches quite well with the bootstrapped threshold.

Table 1 contains more comparisons for $B_{\max} = 10$ and $B_{\max} = 50$.

(a)                                                          (b)

Figure 6: In the offline case: (a) illustration of choosing $b$ by simulation from empirical distribution, (b) comparison of SL $\alpha$ for a range of $b$ values using simulation and Theorem 3.

(a)                                  (b)                                  (c)

Figure 7: Speech data: (a) illustration of background data; (b) choosing $b$ by bootrapping sampling 5000 background data; (b) comparison of SL $\alpha$ for a range of $b$ values using bootstrapping and Theorem 3.

**Online case: accuracy of Theorem 4 and EDD.** We generate 5000 instances to evaluate the accuracy of the threshold for the online case, for various null distributions. Fig. 3 demonstrates that the theoretical threshold is accurate, especially for larger $B_0$.

In the online setting, we compare the EDD of detecting a change-point where the signal is 20 dimensional, and the transition is from a zero-mean Gaussian $\mathcal{N}(0, I_{20})$ to a non-zero mean Gaussian $\mathcal{N}(\mu, I_{20})$ where the post-change mean vector $\mu$ is element-wise equal to a constant mean shift. Fig. 9 compares the EDDs for $B_0 = 20$, using a threshold $4.17$ obtained from simulation, and a threshold $3.73$ obtained from Theorem 4, respectively. Note that when the mean shift is not too weak, the difference between two EDDs is reasonable. Also, EDD decreases as the mean shift increases.

**Dependence on block size.** We evaluate the EDD versus block size under various Gaussian mean shift from $\mathcal{N}(0, I)$. As shown in Figure 10 (a), we fix the Gaussian mean shift to be 0.2 and demonstrate the log (EDD) versus $B$. We see $B = 28$ yields the smallest log (EDD), which is defined as the optimal block size given mean shift equals 0.2. Similarly, we evaluate the specific optimal block size by varying the mean shift from 0.1 to 1.2. See the obtained optimal block size in Figure 10 (b).

Figure 8: In the online case, for a range of ARL values, comparison of $b$ obtained from simulation and from Theorem 4 for various number of blocks $N$.

Figure 9: In the online case, EDD when the post-change distribution is $\mathcal{N}(\mu, I_{20})$, and the post-change mean vector $\mu$ is element-wise equal to a constant mean shift.

# D    Details for real-data experiments.

**CENSREC-1-C Speech dataset.** CENSREC-1-C is a real-world speech dataset in the Speech Resource Consortium (SRC) corpora provided by National Institute of Informatics (NII) [3].

This dataset contains two categories of data:

- Simulated data
  The simulated speech data are constructed by concatenating several utterances spoken by one speaker. Each concatenated sequence is then added with 7 different levels of noise from 8 different environments. So there are totally 56 different noise. Each noise setting contains 104 sequences from 52 males and 52 females speakers.

- Recording data
  The recording data is from two real-noisy environments (in university restaurant and in the vicinity of highway), and with two Signal Noise Ratio (SNR) settings (lower and higher). Ten subjects were employed for recording, and each one has four speech sequence data.

*Experiment Settings.* We will compare our algorithm with the baseline algorithm from [7]. [7] only utilized 10 sequences from "STREET_SNR_HIGH" setting in recording data. Here we will use all the settings in recording data, the SNR level 20db and clean signals from simulated data. See Figure 11 for some examples of the testing data, as well as the statistics computed by our algorithm. For each sequence, we decompose it into several segments. Each segment consists of two types of signals (noise vs speech). Given the reference data from noise, we want to detect the point where the signal changes from noise to speech.

*Evaluation Metrics.* We use Area Under Curve (AUC) to evaluate the computed statistics, like in [7]. Specifically, for each test sequence that consists of two signal distributions, we will mark the points as change-points whose statistics exceed the given threshold. If the distance between detected point

Figure 10: In the online case, when the post-change distribution $Q$ is $\mathcal{N}(\mu, 20)$ and the post-change mean is element-wise equal to a constant mean shift. (a) $\log(\text{EDD})$ versus block size and the optimal block size corresponds to the minimum detection delay; (b) optimal block sizes versus $\mu$.

|  | RH | RL | SH | SL |
|---|---|---|---|---|
| ours | **0.7800** | **0.7282** | **0.6507** | **0.6865** |
| baseline | 0.7503 | 0.6835 | 0.4329 | 0.6432 |

(a) Recording data

|  | C1 | C2 | C3 | C4 | C5 | C6 | C7 | C8 |
|---|---|---|---|---|---|---|---|---|
| ours | **0.9413** | **0.9446** | **0.9236** | **0.9251** | **0.9413** | **0.9446** | **0.9236** | **0.9251** |
| baseline | 0.9138 | 0.9262 | 0.8691 | 0.9128 | 0.9138 | 0.9216 | 0.8691 | 0.9128 |

(b) Simulate clean data

|  | S1 | S2 | S3 | S4 | S5 | S6 | S7 | S8 |
|---|---|---|---|---|---|---|---|---|
| ours | 0.7048 | **0.7160** | **0.7126** | **0.7129** | **0.7094** | **0.7633** | **0.6796** | **0.7145** |
| baseline | **0.7083** | 0.6681 | 0.6490 | 0.7119 | 0.6994 | 0.6815 | 0.6487 | 0.6541 |

(c) Simulated data with SNR=20db

Table 2: AUC results in CENSREC-1-C speech dataset. Simulated data are from 8 noise categories, and with two different noise levels (clean(C) and SNR 20db (S)); Recording data are from RESTAURANT_SNR_HIGH (RH), RESTAURANT_SNR_LOW (RL), STREET_SNR_HIGH (SH) and STREET_SNR_LOW (SL).

and true change-point is within the size of detection window, then we consider it as True Alarm (True Positive). Otherwise it is a False Alarm (False Positive).

We use 10% of the sequences to tune the parameters of both algorithms, and use the rest 90% for reporting AUC. The kernel bandwidth is tuned in $\{0.1d_{med}, 0.5d_{med}, d_{med}, 2d_{med}, 5d_{med}\}$, where $d_{med}$ is the median of pairwise distances of reference data. Block size is fixed to 50, and the number of blocks is simply tuned in $\{10, 20, 30\}$.

*Results.* Table 2 shows the AUC of two algorithms on different background settings. Our algorithm surpasses the baseline on most cases. Both algorithms are performing quite well on the simulated clean data, since the difference between speech signals and background is more significant than the noisy ones. The averaged AUC of our algorithm on all these settings is **.8014**, compared to **.7578** achieved by baseline algorithm. See the ROC curves in Figure 12 for a better comparison.

**HASC dataset.** This data is from *Human Activity Sensing Consortium (HASC) challenge 2011*[4]. Each data consists of human activity information collected by portable three-axis accelerometers. Following the setting in [7], we use the $\ell_2$-norm of 3-dimensional data (i.e., the magnitude of acceleration) as the signals.

We use the 'RealWorldData' from HASC Challenge 2011, which consists of 6 kinds of human activities (walk/jog, stairUp/stairDown, elevatorUp/elevatorDown, escalatorUp/escalatorDown, moving-

(a) RESTAURANT_SNR_HIGH      (b) STREET_SNR_HIGH      (c) Clean-1

(d) RESTAURANT_SNR_LOW      (e) STREET_SNR_LOW      (f) SNR 20db-1

Figure 11: Examples of speech dataset. The red vertical bar shown in the upper part of each figure is the ground truth of change-point; The green vertical bar shown in the lower part is the change-point detected by our algorithm (the point where the statistic exceeds the threshold). We also plot the threshold as a red dash horizontal line in each figure. Once the statistics touch the threshold, we will stop the detection.

Walkway, stay). We make pairs of signal sequences from different activity categories, and remove the sequences which are too short. We finally get 381 sequences. We tune the parameters using the same way as in CENSREC-1-C experiment. The AUC of our algorithm is **.8871**, compared to **.7161** achieved by baseline algorithm, which greatly improved the performance.

Examples of the signals are shown in Figure 13. Some sequences are easy to find the change-point, like Figure 13a, and 13d. Some pairs of the signals are hard to distinguish visually, like Figure 13b and 13c. The examples show that our algorithm can tell the change-point from walk to stairUp/stairDown, or from stairUp/stairDown to escalatorUp/escalatorDown. There are some cases when our algorithm raises false alarm. See Figure 13h. It find a change-point during the activity 'elevatorUp/elevatorDown'. It is reasonable, since this type of action contains the phase from acceleration to uniform motion, and the phase from uniform motion to acceleration.

Figure 12: AUC comparison on speech dataset

Figure 13: Examples of HASC dataset. The markers in this figure are the same as in Figure 11.

## Footnotes

[3] Available from http://research.nii.ac.jp/src/en/CENSREC-1-C.html

[4]http://hasc.jp/hc2011