[Reviews · NeurIPS 2015]

Submitted by Assigned_Reviewer_1

The authors make a nice contribution based on the recent B-test variant instead of MMD_u, with efficient algorithms, additional analysis, and many illustrations and comparisons.

I have the following comments and suggestions:

- in what sense is the change-point detection different from outlier detection problems? (on p.3 the authors mention e.g. one-class SVM which is used also for anomaly detection). It would be good if this could be clarified in the introduction.

- Fig. 2 why are the statistics so different in (b) versus (c)? in both case one has Normal-> Laplacian while the statistics from offline versus online are quite different.

- more attention should be paid to the kernel bandwidth selection. How sensitive are the results in Fig.2 wrt this kernel bandwidth?

Only in section 5 a brief comment is given to [1] on the median trick. Some more explanation is needed in this paper on this bandwidth selection.

Summary: M-statistic for change-point detection based on kernel methods, with characterization of tail probability and efficient on-line and off-line algorithms.

Submitted by Assigned_Reviewer_2

-weak review-

This paper proposes two related M-statistics for change-point detection based on kernel maximum mean discrepancy. It allows to incorporate large amount of background data with only O(n) cost. Furthermore approximations to the significance level / average run length allow for an efficient way to set the change-point detection threshold without recurring to bootstrapping. These approximation are empirically evaluated on real data and the change-point detection performance is evaluated on both real and synthetic data.

The paper is very well written, structured and illustrated. The math seems to be sound and utilizes a highly sophisticated technique based on change-of-measure. The main contributions are: 1. accurate approximations to the significance level in the offline case, and average run length in the online case 2. a closed-form variance estimator

There are a few shortcomings though. First of all, the computational complexity with regard to the time horizon t is hidden in the paper. If I am correct, B_max equals t and they require O(nB^2) for each 2 < B < B_max, which implies a complexity of O(nt^3). This should be mentioned as it limits the possible applications. Secondly it is unclear why RDR is the only baseline algorithm chosen from the nonparametric change-point detection approaches mentioned in the paper.

In summary, this paper contains novel contributions and the authors do a good job in showing when this algorithm is beneficial (large amount of reference data) and that the theoretical change-point detection threshold can be used as an efficient alternative to simulation based approaches.
Summary: In summary, this paper contains novel contributions and the authors do a good job in showing when this algorithm is beneficial (large amount of reference data) and that the theoretical change-point detection threshold can be used as an efficient alternative to simulation based approaches.

Submitted by Assigned_Reviewer_3

The problem addressed in this paper is the determination of the acceptance threshold in an abrupt change detection test using kernels. The proposed solution is two computationally efficient M-statistics adapted for large datasets for both batch and online settings. A theoretical characterization of the tail probability of these statistics is proposed based on a new technique using change of measure. Empirical evidence reported on synthetic and real data illustrates the interest of the proposed approach.

The paper is well-written, well-organized, self contained and represents a significant piece of work on an important and difficult subject: change detection. Literature survey looks rather complete to me. The use of a change-of-measure approach allows deriving a theoretical performance analysis of the proposed test by providing an analytical expression for the variances. This way of doing thinks is new and interesting. Empirical evidence reported is important and convincing.

Minor comments

- Figure 2 is too small and legend (d) is unreadable
Summary: The paper is well-written, well-organized, self contained and represents a significant piece of work on an important and difficult subject: change detection.

Submitted by Assigned_Reviewer_4

This paper proposes two M-statistics, for the offline and online settings, for change-point detection that are based on the kernel MMD B-test statistic for two-sample testing. The B-test statistic is a recently developed alternative to the MMD that is more efficient; it involves taking an average of the MMD over a partitioning of the data into N blocks. Since the blocks are independent, the B-statistic is asymptotically normal so it is simple to obtain thresholds without resorting to bootstrapping. This procedure is modified for the case of change point detection by averaging the MMDs computed from each of N blocks occurring before a potential change point with the block of data of the same size as each N occurring after the potential change point. To determine whether a change-point has occurred somewhere in a block of data, the above procedure is carried out for each potential change point location and a Z-statistic is obtained for each. The M-statistic proposed by the authors is a maximum over the standardized Z-statistics. The authors derive an expression for the variance of the Z-statistics (needed to standardize it) and for the significance threshold of the M-statistic (since the Z-statistics are not averages of independent blocks of data, they are not asymptotically normal like in the case of the B-test) and give a theoretical performance analysis which is also verified numerically. Finally, they apply their method to several real data sets, where it seems to perform slightly better than a baseline method in terms of AUC.

The work is original and significant. Competing methods require bootstrapping, which is much less efficient with larger amounts of data. Furthermore, the theoretical work involved in deriving the variance of the Z-statistic and thresholds for the M-statistic may prove to be useful elsewhere.

The theoretical work is rigorous and seems reasonable. I did not check the proofs in the appendix.

The presentation is generally good, but could probably be improved. I needed to read through section 3 several times before I understood the procedure. The fact that the data are being formed into blocks at two different levels (for each N and each B) make things difficult to understand when described casually in the body of the section. Figure 1 was somewhat helpful but I was still confused. If the procedure were described more formally, e.g. pseudocode, it might be easier to understand. In my opinion, some of the space given to details in sections 4 and 5 could be better used explaining the test procedure better in section 3 (and those details moved to the appendix). Additionally, it would be useful to have more of an explanation of the baseline procedures in section 6 and incorporate some of the results from the appendix.

There are some minor typographical / editing errors in the paper; it should be proofread a few times. Line 111 begins with something that is not a sentence.
Summary: The authors propose a new method for kernel change-point detection that is competitive with existing methods in terms of performance, but is much more efficient because it does not require resorting to bootstrap procedures. The paper is significant, the theoretical work is rigorous and appears to be correct and backed by strong empirical results, and the presentation is generally good though the procedure could probably be explained more clearly and given a more formal treatment.

Author Feedback
Author rebuttal: We would like to thank the reviewers and area chairs for the time and efforts in the review process.

Reviewer 1:
-- Typically, anomaly detection assumes that the anomaly is represented by a small number of samples in a background of normal data. One class SVM makes decision about anomaly for each individual point separately. Change-point detection studies the problem where the change-point causes a change in the underlying distribution of the data, which means that the change will be persistent after that point in time. The decision is made based on data points before and after the change collectively.

-- Fig. 2(b) is the offline M-statistic, and Fig. 2(c) is the online M-statistic. These two statistics, although both derived based on the normalized B-statistic, detect the change-point by different scanning strategies.

In the offline setting, the maximal value of the statistic is obtained via retrospectively scanning through the entire testing data in a reverse direction, which is different from the online setting. Note that in Fig. 2(b), the x-axis represents the block size "B", different from Fig. 2(c) the online setting where the x-axis represents "time". More details about how we scan the testing data in the offline setting can be found in Fig. 1(a). More specifically, we fix one boundary of the test block aligned with the most recent data and adjust the other boundary backwards. In doing this, as shown in Fig. 2(b), the statistic will first increase as the block size "B" increases, due to an increasing number of post-change samples being tested. Then the statistic reaches its peak around the true change-point location. After that, the statistic decreases, resulting from an increasing number of pre-change samples included in the testing data. The way we design the scanning statistic results in the statistic to peak around the location where it is most likely to be a change-point over the entire data.

The online M-statistic, on the other hand, sequentially determines whether a change occurs or not by moving a sliding-window forward. As shown in Fig. 1(b), this time we fix the size of the block and still let the right boundary of the block align with the most recent data. It detects a change when the statistic hits the threshold for the first time and stops (since we design it to be a stopping rule and we will fire an alarm once it stops).

-- We will investigate the kernel bandwidth selection more thoroughly in our journal paper. Here we only briefly explain why we employed the median trick. In a recent study [1], authors justified the use of the median trick in MMD for two-sample test by providing numerical results as well as theoretical insights. They considered a range of bandwidth choices and numerically studied the testing power under each choice, which verified that the median heuristic for a Gaussian kernel maximizes the power in distinguishing two Gaussians with different means. In our study, we adopted the median heuristic in most experiments since we used Gaussian kernel. We add [1] as references in the final version.

[1] A. Ramdas et al. On the decreasing power of kernel and distance based nonparametric hypothesis tests in high dimensions. AAAI 15.

Reviewer 5:
-- Indeed, in the worst scenario, like pointed by the reviewer, the complexity for the OFFLINE case, which requires searching the maximum over all the possible block-sizes, is O(n t^3). However, this calculation does not utilize the structure of the statistics by noticing that many terms in the sum of the Z-statistic are identical and can be taken out of a pre-calculated Gram matrix. So, in practice, we can significantly bring down the complexity by pre-computing the Gram matrix, which has complexity O(n t^2).

For the ONLINE case, however notice that we only choose one block size B_0 regardless of t, and hence the complexity is only O(nB_0^2), and B_0 is typically a small number. We demonstrate numerically that this constant block-size low-complexity approach can still detect the change quickly in many cases.

- RDR (Song et al, 2013) is one of the state-of-art algorithms in this line of research, and it also beats uLSIF (Kawahara and Sugiyama, 2012) and KLIEP (Kawahara and Sugiyama, 2012) on datasets including HASC and a subset of CENSREC. We carried out experiments using the same datasets in a similar setting, which demonstrates the advantage of our M-statistic over RDR. We will also compare the proposed algorithm to more algorithms under various settings in the future work.